# Bacterial Cellulose-Based Polymer Nanocomposites: A Review

**DOI:** 10.3390/polym14214670

**Published:** 2022-11-02

**Authors:** Viktor V. Revin, Elena V. Liyaskina, Marina V. Parchaykina, Tatyana P. Kuzmenko, Irina V. Kurgaeva, Vadim D. Revin, Muhammad Wajid Ullah

**Affiliations:** 1Department of Biotechnology, Biochemistry and Bioengineering, National Research Ogarev Mordovia State University, 430005 Saransk, Russia; 2Faculty of Architecture and Civil Engineering, National Research Ogarev Mordovia State University, 430005 Saransk, Russia; 3Biofuels Institute, School of the Environment and Safety Engineering, Jiangsu University, Zhenjiang 212013, China

**Keywords:** bacterial cellulose, nanocomposites, biopolymers, functional materials, biomedical applications

## Abstract

Bacterial cellulose (BC) is currently one of the most popular environmentally friendly materials with unique structural and physicochemical properties for obtaining various functional materials for a wide range of applications. In this regard, the literature reporting on bacterial nanocellulose has increased exponentially in the past decade. Currently, extensive investigations aim at promoting the manufacturing of BC-based nanocomposites with other components such as nanoparticles, polymers, and biomolecules, and that will enable to develop of a wide range of materials with advanced and novel functionalities. However, the commercial production of such materials is limited by the high cost and low yield of BC, and the lack of highly efficient industrial production technologies as well. Therefore, the present review aimed at studying the current literature data in the field of highly efficient BC production for the purpose of its further usage to obtain polymer nanocomposites. The review highlights the progress in synthesizing BC-based nanocomposites and their applications in biomedical fields, such as wound healing, drug delivery, tissue engineering. Bacterial nanocellulose-based biosensors and adsorbents were introduced herein.

## 1. Introduction

Cellulose is a biopolymer primarily of plant origin that is the most abundant on the earth. Chemically, it is a linear homopolysaccharide consisting of D-glucose residues interconnected by β-1,4-glycosidic bonds. The cellulose macromolecule is composed of thousands of glucose residues. The order in the arrangement of cellulose macromolecules is maintained due to the forces of intermolecular interaction (van der Waals forces) and mainly intramolecular and intermolecular hydrogen bonds. In the structure of cellulose, where each glucose unit has three hydroxyl groups, hydrogen bonds are very important. They affect the physical, physicochemical and chemical properties of the polymer. The bonds give the fibers high strength and insolubility in most solvents. Cellulose macromolecules located in parallel have many intra- and intermolecular bonds leading to the formation of fibrillar-type structures: elementary fibrils, microfibrils, and other larger supramolecular formations [1]. Cellulose macromolecules in fibrils form homogeneous highly ordered crystalline zones (crystallites), which alternate with inhomogeneous, less ordered amorphous zones [2].

Different forms of cellulose (C) are available, such as nano crystals (CNCs), nano fibers (CNFs), nano whiskers (CNWs), microcrystalline cellulose (MCC), and bacterial cellulose (BC) or bacterial nanocellulose (BNC) [3,4,5]. All these forms have been successfully employed in various fields, and can be easily functionalized due to the presence of bulk–OH groups on cellulose backbone to obtain the materials of desired properties [6,7,8]. Despite the same chemical composition, the structure and properties of BC differ significantly from those of plant cellulose. Microorganisms produce extracellular cellulose with high purity, which does not contain lignin, hemicellulose, and pectin; thus, it can reduce the cost of downstream processing. BC does not pollute the environment, since it is biodegradable and completely non-toxic [9]. BNC molecules are arranged strictly parallel to each other forming crystalline microfibrils 100 times thinner than plant cellulose microfibrils. BNC microfibrils combine into nanofibrils, 25–100 nm in diameter, and several micrometers long [10]. The interweaving of fibers forms a porous sponge that absorbing and retaining a great amount of water for a long time. The specific surface area of BC is 200 times greater than that of plant cellulose fibers [11,12]. The moisture content of undried BNC films is about 99% [13]. At the same time, most water is molecularly bound; its molecules are directly bound to the hydroxyl groups of cellulose (only 0.3% of 98% water is in the form of free water) [13]. The BC microfibrillar structure is responsible for most of its properties such as high tensile strength, high degree of polymerization and crystallinity. Due to the correct arrangement of fibers, the BC crystallinity degree can reach 84–90% [14,15,16]. In addition, BC is produced by microorganisms having a short growth cycle, fast metabolism, and high reproduction rate. Therefore, the efficiency of cellulose synthesis is high, and the relative pathway of microbial metabolism is relatively clear, which is more conducive to the cellulose synthesis regulation. An indisputable advantage of BNC is its good formability and the capability of purposeful modification, which offers new opportunities for BNC usage [7,8,17].

Due to its unique properties, BC is a promising material for industry and technology [18]. It has great potential to be used in medicine [4,19,20,21,22] as a biomaterial for tissue engineering [20,23,24], wound dressing [25,26,27,28] and drug delivery systems [5,29,30,31,32], it can be applied in electronics (sensors, energy storage devices, speakers, acoustic membranes) [33], the environmental industry (water purification, filtration and adsorption techniques) [34,35], and the food industry (artificial food, additives, food packaging [36]. The relatively large diversity of cellulose-producing microorganisms, as well as a wide variety of cultivation methods, creates an excellent opportunity to modify and adjust the properties of the material and find new areas of its application.

In contrast to the previously published reviews, this article offers an updated assessment of the latest research findings in the field of highly efficient production of BNC to obtain new nanocomposites with a focus on their biomedical and environmental applications including wound dressings, tissue engineering, drug delivery systems, biosensors, and adsorbents. The review discusses different combinations of BNC with other biopolymers and several biologically active agents (metals, inorganic substances, drugs) to develop novel materials and composites with wide applications in biomedical and biotechnological fields.

## 2. BC Producers

BC is produced by Gram-negative bacteria of the genera *Komagataeibacter* (*Gluconacetobacter*), *Agrobacterium*, *Achromobacter*, *Enterobacter*, *Rhizobium*, *Pseudomonas*, *Salmonella*, *Azotobacter*, and *Alcaligenes*, as well as Gram-positive bacteria *Sarcina ventriculi* and *Rhodococcus* [37,38,39]. The best-known producer of BC is the acetic acid bacterium *Komagataeibacter xylinus* (*Gluconacetobacter xylinus*, *Acetobacter xylinum*, *A. aceti* ssp. *xylinum*, A. *xylinus*). The genus *Komagataeibacter* belongs to the family *Acetobacteraceae*, class *Alphaproteobacteria*, phylum Proteobacteria. The genus *Komagataeibacter* is named after the famous Japanese microbiologist Dr. Kazuo Komagata, Professor at the University of Tokyo, who made a great contribution to the taxonomy of bacteria, especially acetic bacteria. Cellulose-producing bacterial cells are Gram-negative, rod-shaped, and single or pair, some in short chains, their size about 0.4–1.2 μm in width and 1.0–3.0 μm in length (Figure 1A) [39,40]. The colonies of cellulose-synthesizing strains are jelly-like, rounded, and uplift in the center (Figure 1B) [39,41,42].

Numerous recently isolated BC producers belong to the genus *Komagataeibacter* (Table 1). A number of strains were isolated from the Kombucha community: *K. hansenii* GH-1/2008 (VKPM B-10547) [43], *K. xylinus* B-12068 [44], *K. rhaeticus* P 1463 [45]; *K. intermedius* AF2 [46]. The producers *Komagatabacter* (*Gluconacetobacter*) sp. RKY5 [47] and *K. medellinensis* [48] were isolated from vinegar. *K. intermedius* [49]; *G. swingsii*, *G. rhaeticus* [50], *Komagatabacter* (*Gluconacetobacter*) sp. gel_SEA623-2 [51] were isolated from fruit juices; and *Komagatabacter* (*Gluconacetobacter*) sp. F6 [52] was isolated from fruits.

Complete sequences were obtained for the genomes of the following strains: *K. medellinensis* NBRC 3288 [61], *K. nataicola* RZS0111 [62], *K. hansenii* ATCC 53582 [63], *K. xylinus* E25 [64], *K. xylinus* E259, *K. xylinus* CGMCC 2955 [65], *K. xylinus* E26, and *K. xylinus* BCRC 12334 [66]. The DNA G + C content in *Komagataeibacter* varies from 55.8 to 63.4 mol % [67]. Japanese scientists obtained the recombinant *E. coli* bacteria capable of forming BC resulting from the transfer of *G. xylinus* genes [68]. A new stable efficient plasmid-based expression system of recombinant BC in *E. coli* DH5α platform has currently been developed [69].

BC plays an important role for the bacteria themselves [70]. For example, for most aerobes, the maintenance of aerobic conditions can be the reason for its formation. The water-holding capacity of vegetable cellulose reaches 60%, while the water-holding capacity of BC is 100% of its dry weight. Thus, it protects the cells from drying out. It is assumed that BC forms a kind of “framework” protecting cells from foreign substances, heavy metal ions, and from UV radiation effects. In addition, BC is involved in the adhesion of bacterial cells of the genus *Rhizobium* during symbiosis with leguminous plants and *Agrobacterium* during infection, promotes the colonization of plants providing protection from competitors. Cellulose and its derivatives are important components of biofilms and play a significant role in regulating the virulence of plant and human pathogens [70,71].

Inspite of the numerous advantages of BC over plant cellulose, its production is quite expensive. This is chiefly due to the low productivity of bacterial strains, which, as a rule, does not exceed 5 g/L of BC. According to a review by Li et al. [38] the maximum yield of BC did not exceed 20 g/L, which has not yet reached the level of industrial application. Therefore, the screening for bacterial cellulose producing strains is a fundamental strategy for enriching bacterial cellulose producing strain types and highly efficient production of BC. In addition, the BC production can be increased through the development and improvement of technological processes, such as optimization of the culture medium, culture regimes and optimization of cell-free culture systems.

## 3. Cost-Effective Production of Bacterial Cellulose

Although BC is a promising material for biomedicine, industry, and technology, there are unsolved problems of large-scale production of BC and its commercial use related to the high manufacturing cost and low productivity of the producer strains. Therefore, further research is needed to set up a highly efficient production of BC. Recently, many studies have focused on cheap nutrient sources, diverse strains of cellulose-producing microorganisms, and the optimization of their culture conditions and the techniques to produce cost-effective BC [72,73]. The alteration of growth conditions (temperature, pH, oxygen amount, sources of carbon, nitrogen, and their concentrations), different cultivation methods influenced both the quality and quantity of BC yielded [74].

BC production comprises fermentation in static or agitated conditions. The macrostructure morphology of BC varies depending on different culture methods. Under static cultivation, bacteria form cellulose in the form of a film on the medium surface (Figure 2A). Under agitated conditions most strains form cellulose in the form of agglomerates of various shapes and depending on the composition of the medium and mixing modes (Figure 2B,C) [39,75,76,77]. Although dynamic fermentation can improve dissolved oxygen content by shaking or agitation compared to static fermentation, cells are prone to mutation during mechanical agitation of the culture making them lose their ability to produce BC. Cell-free culture, a synthesis without living cells, shows great development prospects [78,79].

An important problem in obtaining BC is the use of expensive nutrient media. About 30% of the total cost of the process is known to be the cost of the nutrient medium [80]. Among the cultivation media, the most frequently used one is a chemically defined medium known as the Hestrin–Schramm (HS) medium. This medium involves some expensive components, such as glucose, peptone, yeast extract, citric acid, and disodium phosphate, resulting in costly production. BC can be produced involving various sugars as a carbon source using both synthetic and non-synthetic media. As a rule, glucose and sucrose are used as carbon sources, although other carbohydrates (fructose, maltose, xylose, glycerol, etc.) can be applied [81]. The efficiency of using a certain carbon source for BC biosynthesis depends primarily on a producer strain [81,82]. Different strains are able to synthesize from 0.5–1.2 to 10–15 g/L of BC from various carbon sources [83,84,85,86,87].

Since cost is a significant limitation in BC production, current efforts are focused on using industrial waste and byproduct streams as a cost-effective substrate for BC production [73,87]. In a review by Kadier et al. industrial wastes are divided into six groups: (1) brewery and beverages industries wastes; (2) agro-industrial wastes; (3) lignocellulosic biorefineries, pulp mills, and sugar industries wastes; (4) textile mills; (5) micro-algae industry wastes; (6) biodiesel industry wastes [73]. Figure 3 shows schematic overview of BC production from different industrial wastes.

There were reported several studies on the feasibility of using different agro-industry wastes in BC production including sisal juice [88], wastes of sugarcane and pineapple [89,90], mango and guava purees [91], fruit [92], rotten fruit culture [80], fruit juices [92,93,94], corn products [95]; sago byproduct [96], corncob and sugarcane bagasse [58], liquid tapioca waste [97], coffee cherry husk [98], date syrup [99], dry olive mill residue [100,101], coconut water [102], oat hull-derived enzymatic hydrolyzates [103,104], enzymatic hydrolysate of wheat straw [105], pineapple and water melon peels [106], citrus peel and pomace [107,108,109], and banana peel [110]. Furthermore, pulp mills and lignocellulosic wastes [111,112]; wastes of biodiesel industry [113]; acetone-butanol-ethanol (ABE) fermentation wastewater [114], and micro-algae biomass industries waste [115] were used as a growth medium for BC production.

Wastes from the alcohol, dairy and sugar industries such as stillage, whey and molasses have been examined by many scientists as alternative substrates for the enhanced production of BC. Thin stillage (TS) contains organic and inorganic compounds, some of which may be valuable fermentation coproducts. Ratanapariyanuch et al. used HPLC to analyze the TS components [116]. They found the major components in wheat thin stillage to be dextrin (8.47–11.65 g/L), glycerol (2.39–7.87 g/L), lactic acid (5.07–7.41 g/L), acetic acid (0.56–2.72 g/L), succinic acid (0.63–0.93 g/L), ethanol (0.23–1.31 g/L), maltotriose (0.14–1.10 g/L), maltose monohydrate (0.03–1.05 g/L), glycerophosphorylcholine (0.91–1.11 g/L) and betaine (0.8–1.03 g/L). The availability of stillage to increase the BC yield was studied [117,118,119]. It is known from the literature that organic acids create a positive effect on BC production. For example, rice wine stillage containing organic acids was used as an additive to the HS medium to increase cellulose yield. The largest amount of BC, 6.31 g/L, is obtained when the HS medium is diluted with stillage by 50%. In the work by Revin et al. in order to reduce the BC cost used the by-products of the dairy and alcohol industry as milk whey and wheat TS [119]. The maximum yield of BC was observed on the TS (6.19 g/L) for 3 days of cultivation under agitated conditions, which is almost 3 times higher than the yield of BC on the HS medium (2.14 g/L). The BC yield on whey was 5.45 g/L. The thinnest BC microfibrils with a higher crystallinity index (82.3 %) were found to form on the stillage (Figure 4).

Whey contains carbohydrates such as lactose (about 70% of dry matter), and small amounts of glucose, galactose, arabinose, and lactulose, as well as proteins, amino acids, vitamins, and organic acids (lactic, citric) [120]. According to literature sources, such organic acids as acetic acid, succinic acid, gluconic acid, citric acid, and malic acid have a positive effect on BC biosynthesis [121,122]. Some researchers have studied the effect of whey on BC formation under static conditions. P. Carreira et al. observed a low level of BC formation on cheese whey—0.08 g/L [123]. Suwanposri et al. obtained BC in the amount of 4.10 g/L on the 7th day of cultivation of *Komagataeibacter* sp. PAP1 using soya bean whey [124].

Molasses is one of the most studied waste products of the sugar industry for BC production [125,126,127,128,129,130]. Molasses is a by-product of the final stage of crystallization in sugar production, and is one of the most economical carbon sources in the microbiological industry. Molasses contains about 80% dry matter, and sucrose comprises about 48%. It is also rich in proteins and organic nitrogen. Abol-Fotouh et al. suggested the preliminary thermal acid treatment of molasses to breakdown the contained sucrose to glucose and fructose [127]. BC is known to be obtained in an amount of 5.3 g/L on a medium with molasses strain *A. xylinum* BPR2001 in a bioreactor after its thermal acid treatment [125]. Revin et al. studied the formation of BC using the *K. sucrofermentans* H-110 strain on a medium with molasses at concentrations of 45 g/L, the sucrose content being 25 g/L under static conditions [130]. The studies have shown the greatest accumulation of BC (2.9 g/L) to occur on the medium with molasses during 5 days of cultivation, which is almost 2 times higher than on the standard HS medium (1.6 g/L). The thickness of BC fibrils formed on a standard HS medium and a medium with molasses is 60–90 nm. The crystallinity degree of BC formed on the medium with molasses was higher than that on the HS medium, and amounted to 83.02%.

## 4. BC-Based Nanocomposites

Recently, BNC has received remarkable attention and has been widely studied due to its excellent structural and physical properties such as high surface area and special surface chemistry, high crystallinity and mechanical strength, hydrophilicity, and excellent biological features (biocompatibility, biodegradability, and non-toxicity). Although BNC exhibits unique features, it lacks the ones like antimicrobial activity, antioxidant activity, electromagnetic properties, and catalytic activity, which are required for its specialized applications [3]. The problem can be solved by modifying the BC surface and creating BNC-based biocomposite materials [3,7,8,131,132]. The cellulose surface modification significantly increased its potential due to its OH group. There are various types of surface modification described with detail in a recent review by Aziz et al. [7]. A recent review by Aditya provides the information on BC functionalization via chemical and physical means to yield nanocomposites and fabricate materials with improved functionalities for the biomedical application, primarily, for vascular and neural applications, wound healing, and bactericidal interfaces [8]. Generally, composites consist of two types of individual materials, namely: the matrix and the reinforcement material, and have a defined interface between them [131,132]. The matrix acts as a scaffold supporting the reinforcement material, while the reinforcements impart the physico-chemical and biological properties to the matrix (Figure 5).

BC composites have been synthesized using numerous materials ranging from natural and synthetic polymers to inorganic nanoparticles and nanomaterials. Shah et al. classified BC composites by the nature of the reinforcement material into organic materials and inorganic materials [132]. These two main classes they further subdivided into BC composites with polymers, NPs, metals, metal oxides, clays and macro-sized solid particles. Until now, many nanomaterials, such as metal nanoparticles (Ag, Au, Pd, Pt, Ni) and metal oxides nanoparticles (ZnO, CuO, MgO, FeO, TiO_2_, Al_2_O_3_, CeO), mineral nanomaterials (SiO_2_, CaCO_3_, montmorillonite) and carbonaceous nanomaterials (grapheme, carbon nanotube) have been placed into nanocellulose matrices to prepare BC nanocomposites (Figure 6). There have also been obtained BC biocomposites with biopolymers such as chitosan (Ch), alginate (ALG), hyaluronic acid (HA), starch, gelatin (GT), collagen, keratin, polylactic acid (PLA), polyhydroxyalkanoate (PHA) and synthetic polymers such as polyvinyl alcohol (PVA), polyaniline (PANI), poly-2-aminoethyl-methacrylate (PAEM) [133]. A series of novel polysaccharide-based biocomposites was obtained by impregnation of BC produced by *K. rhaeticus* with the solutions of negatively charged polysaccharides such as hyaluronan, sodium alginate, or carrageenan, and subsequently with positively charged chitosan [134]. In addition, BC composites with biomolecules such as antibiotics, enzymes, hormones, peptides, amino acids and cells were obtained [3,4,5,6,7,8,9].

There are four main methods compounds can be loaded in the cellulose matrix, they can be loaded during BC synthesis; in post-synthesis via saturation; chemical modification once the cellulose has been processed and purified; and finally, through the genetic manipulation of the cellulose-producing organism [21]. A review by Shah et al. and Mbituyimana et al. summarized the strategies for BC-based composites with improved properties [132,135]. The synthesis of BC-based composites has adopted different strategies depending on application purposes. The most common methods used for BC composite preparation are: in situ approach, ex situ approach, and the synthesis of a BC composite from a BC solution [132]. The in situ approach involves reinforcement substances added into the culture medium during BC synthesis, which finally becomes a part of the produced BC hydrogel. In the ex-situ method, composites are produced by adding or impregnating reinforcement materials into a synthesized polymer. In the first case, a reinforcing element is introduced into the cultivation medium of the producer, and the composite is obtained in the process of BC biosynthesis. Thus, Saibuatong and Phisalaphong synthesized BC-Aloe vera composites in the form of films under static conditions of producer cultivation by adding different amounts of Aloe vera gel to the synthetic medium [136]. Park et al. obtained a 3D BC-based scaffold by cultivating *G. xylinus* in a culture medium containing carbon nanotubes [137]. The synthesized scaffolds were implanted into the mouse skull for bone tissue regeneration. Gao et al. used 6-carboxyfluorescein-modified glucose (6CF-Glc) as a carbon source to modify BC by fermentation of the bacterium *K. sucrofermentans* [138]. In another recent study, Wan et al. [139] prepared a composite of BC and silver nanowires (AgNW) using in situ biosynthesis. The synthesized BC/AgNW dressings showed better release of Ag+ and a high ability to improve cell proliferation, skin regeneration and the formation of epithelial tissue according to the in vivo wound healing test. Thus, in situ BC-based composites have significantly improved mechanical properties, crystallinity, and thermal stability [140]. However, the use of static cultivation conditions is not always possible to obtain biocomposites using the in situ method, since the particles are in suspension for a short period of time, and a film is formed on the medium surface. Therefore, the cultivation of the producer under dynamic conditions is often used to obtain. The in situ method limitation are also the toxicity and an antibacterial effect of some metals and metal oxides, such as Ag, ZnO, TiO_2_, and antibiotics, which inhibit the growth of microorganisms.When using the ex situ method, reinforcing materials are introduced into BC after its biosynthesis. Soluble substances and solid nanoparticles easily penetrate a porous cellulose matrix. The interaction can be physical as a result of absorption and due to the formation of hydrogen bonds. A large number of composites with polymers, inorganic materials, metals, and metal oxides have been obtained using this method [16,17,18,19]. The method is often used for the production of medical biocomposites. Thus, Fatima et al. obtained an ex situ BC-based composite with antimicrobial activity by introducing bactericidal plant extracts into its three-dimensional matrix [141]. Ul-Islam et al. using an ex situ method developed a high tensile strength BC-Aloe vera gel composite for potential environmental and medical applications [142].

### 4.1. BC-Based Nanocomposites for Biomedical Applications

Recently, BC-based nanocomposites have greatly advanced in biomedical applications, such as wound healing dressings, tissue engineering, drug delivery, and cancer treatment. However, most of these materials have restrictions, for example, a lack of antibacterial activity and low mechanical properties [135]. Recently, several reviews on BC materials for biomedical applications have been reported [18,19,20,21,22,143,144,145,146,147], and a summary of various biomedical applications of BC and BC-based composites developed through different strategies is provided in Table 2.

#### 4.1.1. Wound Dressings

At present, there are various wound dressings that can protect a wound from further injuries, or isolate the external environment in wound treatment [25,26,27,28,185,186,187,188,189,190]. An ideal wound dressing should facilitate healing, maintain moist environment, absorb exudates, support angiogenesis, allow gaseous exchange, prevent microbial infections, be comfortable, and cost-effective. It should be non-toxic, non-allergenic, non-adherent, and should be easy to remove without trauma [26,185]. BC has attractive features in wound healing, including its good flexibility, strong water holding capacity, biocompatibility, vapor permeability, elasticity, and non-toxicity [190,191,192]. The microfibrillar structure of BC serves as a flexible 3D scaffold that can serve as a physical barrier against pathogens contributing to cell attachment and tissue granulation. In addition, BC can be modified to meet all the necessary functional requirements as a wound dressing [187].

Recently, a series of commercial medical materials based on BC has been obtained such as Biofill^®^ (Curitiba, Brazil) and Bioprocess^®^ (Curitiba, Brazil) for the therapy of burns and ulcers, Gengiflex^®^ (Curitiba, Brazil) to treat periodontal diseases, Dermafill^®^ (Londrina, Brazil) for effective wound-healing burns and ulcers, Membracel^®^ (Curitiba, Brazil) for venous leg ulcers and lacerations, xCell^®^ (New York, NY, USA) to treat chronic wounds, EpiProtect^®^ (Royal Wootton Bassett, UK) for burn wounds, and Nanoskin^®^ (São Carlos, Brazil) (the antimicrobial BC product incorporated with silver ions) [20,193]. By the product form and a method of administration, wound dressings are divided into: films, hydrogels, sponges, foams, fiber scaffolds, bandages (Figure 7) [185].

BC has many excellent properties for wound healing, and acts as an effective physical barrier for a bacterial infection, but the lack of antibacterial activity limits its application in wound dressings. The BC functionalization by adding antimicrobial agents can solve the problem. A review by Zheng et al. considered several wound dressings of nanocellulose with inorganic nanomaterials (metal/metal oxides, carbon-based nanomaterials, nanosilicates), organic antimicrobials (natural polymers, bioactive materials, synthetic materials), and antibiotics [188]. The most acceptable form of new wound dressings is BC nanomaterials with nanoscale inorganic particles. For this purpose, silver nanoparticles are included [194,195,196,197], and they have antimicrobial, anti-inflammatory and healing. Silver nanoparticles (AgNP) incorporated into the BC matrix imparts antibacterial properties to the composite through the release of silver ions affecting DNA replication, the breakdown of the cell membrane, and the release of reactive oxygen species [197]. Pal et al. developed the Ag/BC nanocomposite for wound healing with antibacterial activity against *E. coli* using a UV photochemical reduction process [195]. Wan et al. developed a novel composite for wound-dressing by dispersing silver nanowires in BC [139]. Metal oxides such as TiO_2_, CuO, CeO_2_, ZnO, etc. also exhibit antibacterial activity and promote wound healing [154,155,156,157,198,199,200,201]. Therefore, the combination of BC composites with TiO_2_ and ZnO also displayed excellent antibacterial properties and had cellular adhesion and proliferation of fibroblast cells, thereby improving the wound-healing capability [154,156]. Similarly, Khalid et al. developed a BC–ZnO nanocomposite for healing burn wounds. The composite succeeded in killing about 90%, 87.4%, 94.3%, and 90.9% of *E. coli, Pseudomonas aeruginosa, Staphylococcus aureus*, and *Citrobacterfreundii*, respectively [200]. A design of new nanocomposites of BC and betulin diphosphate (BDP) pre-impregnated into the surface of zinc oxide nanoparticles (ZnO NPs) to produce wound dressings was suggested [201]. The effective wound healing with BC-ZnO NPs-BDP nanocomposites can be explained by the synergistic effect of all nanocomposite components, which regulate oxygenation and microcirculation reducing hypoxia and an oxidative stress in a burn wound. Another direction of BC functionalization is the creation of composites with other biopolymers, such as chitosan, alginate, hyaluronic acid, collagen, etc. Chitosan (CS) is one of the most important biopolymers for wound dressings [153,161,202,203,204,205]. CS in a biocomposite has an antibacterial effect against *E. coli* and *S. aureus*, exhibits a wound healing effect and accelerates epithelialization. CS molecules easily penetrate into the BC matrix resulting in the formation of hydrogen bonds between the OH groups of BC and the NH groups of CS. In this case, the mechanical strength of the composite increases. Cacicedo et al. developed a ciprofloxacin-loaded CS-BC patch showing cytocompatibility with human fibroblasts and high antibacterial activity against *P. aeruginosa* and *S. aureus* for potential wound healing [204]. Cazón et al. developed BC films combined with CS and polyvinyl alcohol [205]. Volova et al. developed a hybrid wound dressings using two biomaterials: BC and copolymer of 3-hydroxybutyric and 4-hydroxybutyric acids—a biodegradable polymer of microbial origin [206]. Mohamad et al. developed a hydrogel for burn wounds based on BC and acrylic acid with fibroblasts and keratinocytes added [207].

The antibacterial activity of BC-based wound dressings is often achieved by adding antibiotics. The most commonly used antibiotics are tetracycline hydrochloride, amoxicillin, ciprofloxacin, ceftriaxone, etc. [208,209,210,211]. Junka et al. have shown BC saturated with gentamycin significantly to reduces the level of biofilm-forming bone pathogens, namely *S. aureus* and *P. aeruginosa* [208]. BC composite materials containing amikacin and ceftriaxone were prepared by immersing dried BC films in antibiotic solutions of various concentrations. Moreover, the composites have obvious antibacterial activity against *E. coli, P. aeruginosa, S. pneumoniae and S. aureus*, so they are supposed to be used as wound dressings [196]. The composites of BC with tetracycline hydrochloride were obtained and characterized by other authors [209]. The composites exhibited excellent antibacterial activity and good biocompatibility as well as controlled the antibiotic release. Vancomycin and ciprofloxacin can be incorporated into BNC or modified BNC to confer biological activity in wound dressings and tissue engineering scaffolds [211]. Volova et al. obtained BC composites with silver nanoparticles (BC/AgNPs) and antibacterial drugs (chlorhexidine, baneocin, cefotaxime, and doripenem) with antibacterial activity against *E. coli* and *S. aureus*, and investigated the structure, physicochemical, and mechanical properties of the composites [212]. Revin et al. developed the biocomposite materials for medical purposes with antibacterial, regenerative, and hemostatic properties based on BC in the form of aerogels, hydrogels, film forms, and fusidic acid (FA) [213,214]. FA is an antibiotic, with high antibacterial activity against *S. aureus*, including the MRSA strains [215]. The inhibitory effect of FA on the biofilm formation and the expression of α-toxin was reported [216,217]. BNC- FA biocomposite films with excellent antibacterial activity against *S. aureus* were obtained by immersing dried BNC films in a solution of the antibiotics of various concentrations for 1–24 h [213] (Figure 8).

The use of aerogels for the production of biomaterials has started relatively recently [218]. In the past decade, aerogels have attracted great interest due to their special properties (large porosity, high internal surface, controlled pore diameter, and three-dimensional interconnected structure). Biopolymer-based aerogels additionally provide excellent cytocompatibility, biocompatibility, and biodegradability, and can be successfully used in biomedicine for targeted drug delivery, tissue engineering, and antibacterial materials [218,219]. Revin et al. for the first time obtained new biocomposites with antibacterial properties based on native BC and sodium fusidate (NBC/SF) and TEMPO oxidized BC and sodium fusidate (OBC/SF) in the form of aerogels by incorporating sodium fusidate (SF) into hydrogel native BC and oxidized BC [214] (Figure 9). The antibacterial activity of the resulting aerogels was studied by the disk diffusion method. The biocomposites with sodium fusidate BC/SF and OBC/SF show high antibacterial activity, their *S. aureus* inhibition zone diameters are 28 and 27 mm, respectively (Figure 9D). The present study clearly illustrates that the resulting aerogels exhibit excellent antibacterial activity against *S. aureus*. Despite the small difference in antibacterial activity, OBC/SF aerogels had greater mechanical strength than BC/SF aerogels.

Malheiros et al. immobilized antimicrobial peptides of *Lactobacillus sakei* on BC [220]. Bayazidi et al. obtained a material with antibacterial activity by immobilizing lysozyme on BC [221]. Gupta et al. obtained BC nanocomposite wound dressings with curcumin, which has antimicrobial, antioxidant, antitumor, and wound healing effects [222].

#### 4.1.2. Tissue Engineering

Recently, BC has attracted much attention in tissue engineering due to their unique properties for tissue regeneration as scaffolds [22,23]. 3D BC scaffolds provide an almost ideal environment for cell growth and tissue development, unlike 2D materials, where only superficial growth occurs. Therefore, BC 3D scaffolds become potential candidates for being used in tissue engineering and regenerative medicine. The porous structure of BC enables massive transfer of nutrients and oxygen, supporting cell survival. It can support the growth of endothelial, smooth muscle cells, chondrocytes, and cause no toxic effects. From the cellular point of view, an important feature of BC is the structure of its nanofibrils, which resembles the structure of extracellular matrix components, namely collagen [223]. BC and collagen have the same diameter (less than 0.1 µm), both are polymers functioning, primarily, as mechanical support structures.

##### Cartilage Tissue Engineering

The regeneration problem of articular cartilage damage is important because of the limited ability of self-repair. The cartilage repair requires biomaterials with good porosity and a certain pore size, where chondrocytes can penetrate and proliferate to produce their extracellular matrix. BC is a suitable scaffold for cartilage tissue engineering due to mechanical strength and biocompatibility [224]. Svenson et al. reported BC potential to proliferate cartilage chondrocyte cells [225]. However, native BC has a small pore size (~0.02–10 µm) and therefore cannot provide the penetration of cartilage cells. Therefore, the porosity of the BC scaffold requires improvement. Several methods were implemented to enhance the pore size and interconnectivity of BC scaffolds. To increase the pore size of BC scaffolds, some authors developed BC scaffolds based on agarose particles with a pore size 300–500 µm [226,227] and 150–300 µm [224] followed by removal of progen particles (agarose), for example, by autoclaving [226] or extrusion [224]. Xun et al. developed a macroporous scaffold with a pore size of 200 µm by using the freeze-drying technique for BC suspension followed by crosslinking [228]. Methacrylate gelatin–BC hydrogels with the pores 200–10 µm in size were developed using photo polymerization [229]. Horbert et al. developed a novel technique for 3D-laser perforation of BC seeded with chondrocyte cells [230]. Yang et al. suggested preparing 3D structures simulating intervertebral discs [231]. In conclusion, the improvement of BC scaffold porosity makes it appropriate for cartilage regeneration.

##### Bone Tissue Engineering

Attention to the creation of artificial composite materials similar to natural bone tissue is growing rapidly, and a major problem is to obtain a composite that would be as close as possible in structure and properties to its natural counterpart. It is known that bones are composed of bone tissue cells such as osteoblasts, osteocytes, and osteoclasts. And the matrix of bone tissue mainly consists of collagen and hydroxyapatite.

Recently, a number of BC-based composites have been developed for bone tissue regeneration [22,232,233,234,235]. Due to its strength characteristics (the tensile strength of the BC gel film is ~10 GPa), BC can serve as a promising basis for a bone precursor. BC can serve as a scaffold for proliferation and potential differentiation of mesenchymal stem cells into osteocytes and chondrocytes [236]. Sundberg et al. developed macroporous mineralized BNC scaffolds coated with calcium phosphate. [237]. A number of scientists have also used osteogenic growth peptide for bone tissue engineering [233,238,239]. A composite based on BC and hydroxyapatite (HA) nanocrystals, biocompatible with living organisms, is considered promising as a bone tissue precursor. The BC composite with hydroxyapatite mimics the intercellular substance of normal bone tissue. It plays the role of a barrier preventing loose connective tissue from replacing the lost or destroyed fragment of the skeleton. At the same time, bone tissue cells—osteocytes—can be grafted on it, and they will multiply there. Tazi et al. developed BC-HA scaffolds to improve osteoblast adhesion and growth [232]. Ran et al. developed an organic–inorganic multicomponent composite using BC, gelatin, and HA combination to provide better mechanical properties [240].

Moreover, 3D printing is a promising method for bone tissue engineering. It is used to synthesize composite scaffolds with controlled porosity, mechanical strength, and shape to facilitate cell growth and regeneration [241,242]. For example, Cakmak et al. developed a 3D printed BC/polycaprolactone/gelatin/hydroxyapatite composite scaffold for bone tissue engineering [243]. A review by Khan et al. demonstrated that nanocellulose does not only serve as the matrix for the deposition of different materials to develop bone substitutes but also functions as a drug carrier to treat bone diseases [22]. Hernández et al. have recently developed a BC-based composite scaffold with carbon nanotubes to improve its mechanical properties [234].

##### Soft Tissue Engineering

According to the literature data, BC-based composites have great potential as biomaterials for soft tissues, such as blood vessels, adipose tissue, nerves, the liver, and skin [22,23,244,245,246,247,248,249]. Recent studies have shown BC to be a promising material for vascular tissue engineering with attractive properties such as biocompatibility, high burst pressure, and ultrafine fibrous collagen-like structure [145,249,250]. By taking the advantage of producing BC of different shapes, Schumann et al. and Leitao et al. developed small-caliber vascular grafts [245,251]. So, to replace small arterial, Schumann et al. developed a small-caliber vascular graft, 1.0–3.7 mm in diameter, 5.0–10.0 mm long, and its wall-thickness being 0.7 mm [245]. Leitao et al. developed a simple, cost-effective method for producing small-caliber BC graft vascular prostheses using the capillary drying, shaping, and freeze-drying [251]. Recently, BC-based composites have been shown to be promising scaffold candidates with good biocompatibility and high transmission properties for the corneal stroma [252,253,254]. Moreover, BC-based composites are considered an attractive material for creating neuronal implants due to their high biocompatibility, flexibility, and ability to register nerve signals [255]. So, Yang et al. developed Au–BC microarrays for neural interfaces [256]. Hou et al. developed a biodegradable, biocompatible scaffold with good mechanical properties based on oxidized BC [257]. The current preclinical and clinical studies have confirmed the prospects for studying BNC in neurosurgery as implants for closing the defects in the dura mater in the spinal cord and its meninges pathologies [258].

#### 4.1.3. Drug Delivery System

Drug delivery systems refer to the advanced technologies used for targeted delivery and/or controlled release of therapeutic drugs [31]. In the past few decades, drug delivery systems received much attention, since they offer potential benefits, such as reducing side effects, improving therapeutic effects, and possible reduction of drug doses [32,259]. There are three key factors required in an effective drug delivery system, including drug carriers, drug-loading ratio, and controlled release rate [260].

In recent years, natural polysaccharides have been considered as the ideal candidates for novel drug delivery systems due to their good biocompatibility, biodegradation, low immunogenicity, renewable source and easy modification [29]. These natural polymers are widely used in designing nanocarriers, which find a wide application in therapeutics, diagnostics, delivery and protection of bioactive compounds or drugs. Recently, interesting reviews were published in the Journal ‘Polymers’ characterizing the composite materials used in drug delivery systems [5,29,30]. The review by Qiu et al. introduced a series of polysaccharide-based nanodrug delivery systems such as nanoparticles, nanoliposomes, nanomicelles, nanoemulsions and nanohydrogels for diabetes treatment [29]. The review Huo et al. represented the typical drug release behaviors and the drug release mechanisms of nanocellulose-based composite materials, and considered the potential application of these composites [30]. Due to its adaptable surface chemistry, high surface area, biocompatibility, and biodegradability, nanocellulose-based composite materials can be further transformed into drug delivery carriers [30]. The review by Lunardi et al. reported various methods for modifying and functionalizing nanocellulose to obtain nanocarriers in drug delivery systems [5].

Generally, the most common method of loading drugs in BC membranes is via immersion in a drug solution, usually following lyophilisation to provide maximum drug absorption [21]. The most common drugs to be incorporated into BC are anti-inflammatory drugs, such as ibuprofen and diclofenac [261], and antimicrobial drugs [209]. Due to its unique characteristics, BC has been shown to be a promising biomaterial for cancer treatment [262,263,264,265]. For example, Cacicedo et al. combined a BC hydrogel and nanostructured lipid carriers to use as an implant for local drug delivery in cancer therapy using doxorubicin as a model drug [264]. Zhang et al. developed BC-based composites with Fe_3_O_4_ magnetic nanoparticles coated with doxorubicin and hematoporphyrin monomethyl ether, and additionally conjugated with folic acid for breast cancer therapy [265]. Nanocellulose-based scaffolds are also being used as useful tools for cancer diagnosis. BC can be used as a drug delivery system to treat diseases. For example, it has been used to control curcumin delivery to improve tissue granulation in addition to its antifungal, antitumor, antibacterial, and antioxidant properties [168]. BC has been used to deliver lidocaine to promote tissue repair in third-degree burns in rats [266]. Moreover, BC has been used to deliver antibacterial and antiseptic agents [267]. Luo et al. developed a BC composite with graphene oxide with controlled release of ibuprofen [173]. Ahmad et al. developed a polyacrylic acid hydrogel grafted with BC for oral protein delivery [268].

### 4.2. Biosensors

Biosensors are small devices with a biologically active element. They quantify (or semi-quantify) a biological or chemical analyte by generating a measurable signal proportional to its concentration [269]. Biosensors have provoked great interest in recent years. They are considered as powerful emerging tools for detecting various biomarkers for both healthcare and environmental monitoring [270,271]. Biosensors are generally applied in different fields: biomedical, environmental, and allow to monitor specific disease biomarkers in body fluids (blood, urine, saliva, and sweat) [272], and detect microorganisms [270] and pollutants [273] in the environment.

A biosensor is mainly made up of three elements including a biologically active element immobilized on a convenient substrate such as cellulose, a transducer, and a signal processor (Figure 10). A biologically active element could be an enzyme, antibody, protein, whole cell, or DNA. Biosensors can be classified based on a type of transducers and operating principles into optical, acoustic, electrochemical, and piezoelectric. Optical biosensors take the advantage of optical characteristics such as absorbance, fluorescence and chemiluminescence. For instance, a fluorescence biosensor based on BC and nitrogen-doped carbon quantum dots (N-CDs) was fabricated for the first time by Lv et al. to determine Fe^+3^ ions in a liquid medium [274]. Acoustic biosensors based on piezoelectric crystals operate by detecting the binding of the analyte (target) by its modulation of the crystal oscillation frequency. In electrochemical biosensors, the changes in the electrical properties are used as a measuring parameter. For instance, Zhang et al. prepared a BC biosensor for H_2_O_2_ detection [275].

When developing biosensors, it is of primary importance to ensure high biocatalytic activity, sensitivity, selectivity, environmental friendliness, and low cost. BC is a promising material for creating biosensors, since it is an environmentally friendly natural three-dimensional nanostructure and characterized by high absorption capacity, large surface area, high crystallinity, mechanical strength, and can be easily modified and functionalized with nanoparticles, carbon nanotubes, metal oxides, conductive materials, and biomolecules [276]. BC has a great potential for developing cytosensors due to its various unique properties including biocompatibility. The review by Kamel and Khattab presents the current techniques for the preparation and modification of cellulose substrates as biosensors [269] The review by Torres considered the recent advances regarding the development, production and applications of new BC-based biosensors [276]. To increase the sensitivity of the cellulose surface, novel modifications can be made using conductive materials such as gold nanoparticles (AuNP), carbon nanotubes, graphene oxide (GO), and conductive polymers. Different materials have been used to prepare BC-based biosensors. Gold (Au), palladium (Pd) and platinum (Pt) nanoparticles, and titanium dioxide (TiO_2_), ferrous oxide (FeO) and Zinc oxide (ZnO) have been used to increase the electric and magnetic conductivity of native BC [275,277,278,279,280,281,282]. Polystyrene sulphonate and polyaniline have also been used to increase BC conductivity to fabricate biosensors [177,283,284,285]. Enzymes such as laccase and haem proteins including glucose oxidase and horseradish were immobilized in BC-based networks to prepare electrochemical biosensors to detect H_2_O_2_, hydroquinone, dopamine and glucose, among others [275,278,279,281,282]. BC-based biosensors are prepared by adding a second phase into the BC network [277,286,287], either by modifying the culture medium during cellulose synthesis [274,288] or by incorporating the second phase after obtaining the BC network [289,290]. A different approach is used when the BC network is first destroyed and then combined with another material to prepare BC-based biosensors [275,291]. Future investigations should place special emphasis on overcoming the current limitations related to the immobilization of enzymes on the BC surface in order to expand the potential application of BC biosensors. The applications will include the development of biosensors for detecting biomarkers produced from cells or tissues resulted from diseases and disorders.

### 4.3. Adsorbents

In recent years, adsorption has been used as an effective strategy for separating contaminants due to the fact that an adsorbent can be recovered, reused and recycled, and the method is considered the best, however, the high cost of sorbents limits their use [292]. The use of adsorbates of natural origin is promising, since they are harmless to the environment and human health. Among them, natural fibers such as cellulose have been extensively studied. Shi et al. published a review article summarized the recent progress of adsorbents produced by modification and functionalization of cellulose and cellulose-based nanocomposites to remove heavy metals ions and organic pollutants [34]. An exciting review by Salama et al. presents the latest research results on nanocellulose-based materials for wastewater treatment including adsorption, absorption, flocculation, photocatalytic decomposition, disinfection, etc., and discusses various approaches to their chemical modification [293].

BNC attracts special attention due to the following characteristics: a large surface area, high adsorption capacity, nanoscale structure, high reactivity due to the presence of hydroxyl groups on the surface, which allows it to be chemically modified to interact with various pollutants depending on its nature [294]. Other important BNC properties are biodegradability, high mechanical properties, and low density [295]. BC has many advantages to be used as an adsorbent, including high surface area and density of functional groups [296]. Numerous hydroxyl groups (or others if chemically modified) on nanocellulose result in higher adsorption capacity [297,298,299,300,301]. In addition, nanocellulose-based materials are completely biodegradable, which ensures their biological application without side effects [302,303]. BC serves as a matrix for immobilizing catalysts, enzymes, and other sensory materials, both to detect environmental pollutants, and also for the decomposition of various wastes, for example, waste from the textile industry, which can later be used as a carbon source and biotransformed into valuable products.

BC has the ability to function as a matrix for incorporating various molecules or inorganic particles. Therefore, BC composites can be used for wastewater treatment from heavy metals [304,305,306,307]. A number of BNC-based adsorbents have been obtained for removing hazardous metals. For example, Shoukat et al. developed a nanocomposite based on BC and titanium oxide (TiO_2_) [305]. The nanocomposite was evaluated as an adsorbent to remove lead (Pb) from an aqueous solution. The TiO_2_-BC nanocomposite removes Pb at a concentration of 100 mg/L with a removal efficiency of more than 90% in 120 min at pH 7 and at a room temperature. The adsorbent proved to be effective, stable and reusable for removing lead from environmental water samples. Biocomposite aerogels based on BC and polyaniline can be used to remove hexavalent chromium [306]. A comparative study on the efficiency of mercury removal from wastewater using BC membranes and their oxidized analogue was carried out [35]. The results obtained showed the modification of BC by oxidation to improve the mercury removal capacity, making the modified membranes an excellent material for mercury removal from wastewater. Mensah et al. developed a composite based on BC and graphene oxide as an efficient and environmentally friendly adsorbent for removing metal ions, especially Pb^2^+, from an aqueous system [307].

Another important problem is the purification of water from fluorine. The high content of fluorine in water is an urgent issue all over the world. But despite the growing concern about water pollution, effective technologies for removing fluoride have not yet been developed. Among various physical and chemical methods for removing fluorine from water, adsorption is a preferable one due to its simplicity, relatively low cost, and industrial scalability. Recently, a number of inorganic [308,309,310,311,312] and organic [313,314,315] adsorbents have been obtained to remove fluorine from water. Revin et al. developed a new biocomposite material with a high sorption capacity for fluorine ions (80.1 mg/g) based on BC and nanosized aluminum oxide films chemically immobilized on its surface using ALD technology (Figure 11) [130].

Nanocellulose-based materials can also be used to treat wastewater contaminated with hazardous organic pollutants, including dyes, pharmaceutical compounds, and petroleum products [292,316,317,318,319,320]. For example, Wang et al. reported a new superabsorbent aerogel based on BC and graphene oxide, which showed excellent absorption property of organic liquids [316]. Cellulose composite aerogels can be used for oil sorption [318,319]. The development and synthesis of new materials is vital to the removal of new contaminants, such as pharmaceuticals, from polluted water. Ieamviteevanich et al. developed a magnetic carbon nanofiber derived from BC to remove diclofenac from water [320]. Diclofenac is one of the non-steroidal anti-inflammatory drugs widely used to treat acute and chronic pain in humans and animals. Conventional processes used in wastewater treatment are not sufficient to remove diclofenac from water. The efficiency of these processes is less than 20%. Therefore, alternative methods for removing diclofenac from water have been explored. The maximum adsorption capacity of magnetic carbon nanofiber is 67 mg/g. The results of the study showed it can effectively adsorb diclofenac from water with further removal by magnetic separation. Photocalysis can be considered as another promising method for the degradation of organic pollutants and dyes in wastewater [293]. Although nanocellulose alone exhibits the limited photocatalytic activity in the visible and UV spectra, photocatalysts such as metal oxides ZnO [321] and TiO_2_ [322] can be added to enhance the photocatalytic activity. Nanocellulose can also be used for antimicrobial filtration. So BC membranes were examined for removal of *E. coli* from a sanitary effluent [323].

## 5. Conclusions and Future Trends

In summary, BNC is an important natural biopolymer for various applications due to its unique ultrafine network nanostructure and the properties such as biological compatibility, high mechanical strength, high purity, and high sorption capacity. In recent years, there has been a sharp increase in publications on BNC-based biocomposites obtained for biomedicine, industry, and technology. New biocomposites for wound dressings, tissue engineering and drug delivery systems have been obtained, including those used for vascular tissue engineering and neural implant materials, for cancer and diabetes treatment. Moreover, new methods for obtaining BNC-based biocomposites were developed to create various biosensors and adsorbents, which enabled to solve a number of important medical and environmental problems. However, the commercial production of such materials is limited by high manufacturing cost and low yield. Therefore, further researches are needed to obtain new highly productive strains of bacteria using genetic and metabolic engineering, develop new cost-effective culture media with industrial waste and byproduct streams, as well as the optimization of cells and cell-free culture systems and equipment improvement. In addition, it is important to focus on the development of new approaches for the modification and functionalization of BNC-based materials.The possible combination of such approaches in the future will provide an opportunity to obtain new unique materials able to change life quality and the human environment.

## Figures and Tables

**Figure 1 polymers-14-04670-f001:**
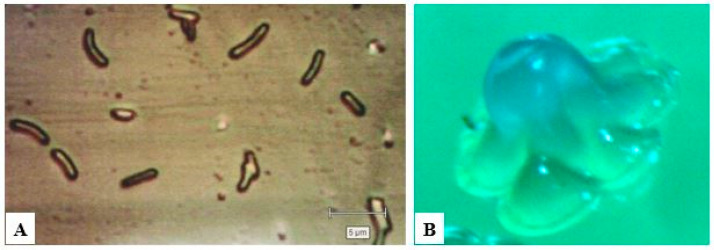
Cell morphology (**A**) (×1500) and colonies (**B**) (×100) *K. sucrofermentans* H-110.

**Figure 2 polymers-14-04670-f002:**
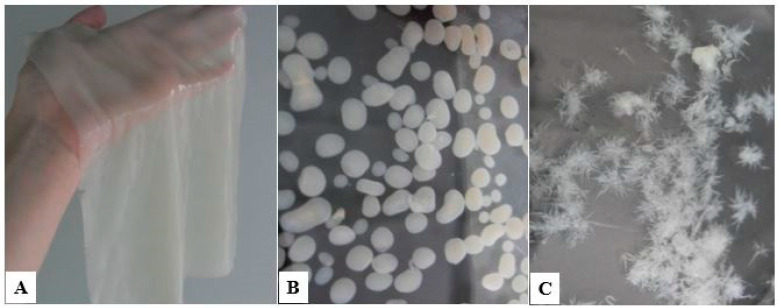
BC gel film (**A**) obtained in static conditions, and BC agglomerates of various shapes (**B**,**C**) formed in agitated culture conditions.

**Figure 3 polymers-14-04670-f003:**
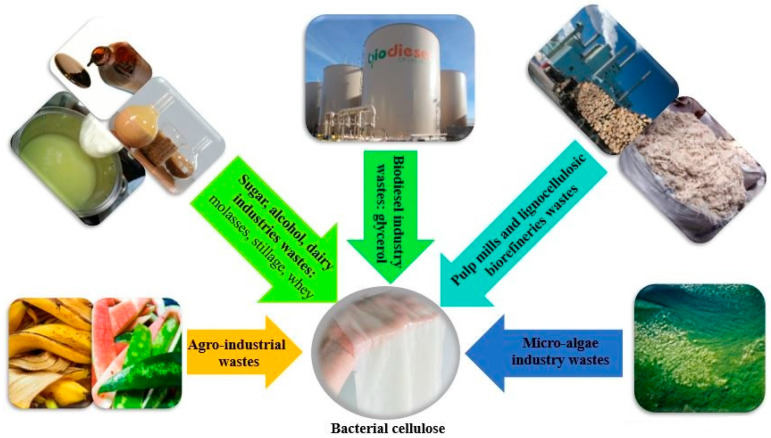
Schematic overview of bacterial cellulose production from different industrial wastes.

**Figure 4 polymers-14-04670-f004:**
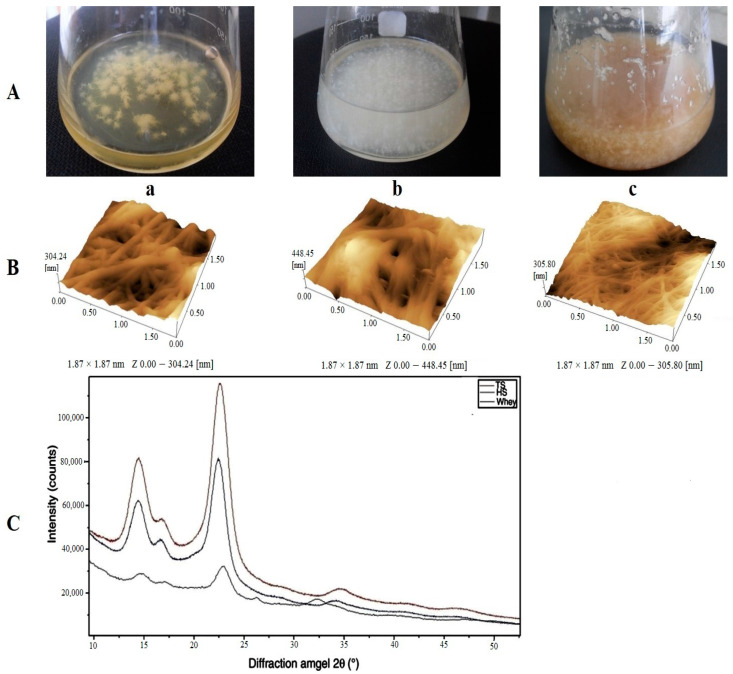
BC produced by *K. sucrofermentans* B-11267 in agitated culture conditions using HS medium (a), whey (b) and thin stillage (TS) (c) (**A**); AFM image of the cellulose microfibrils (**B**), and XRD patterns of BC (**C**). Adapted with permission from Ref. [119].

**Figure 5 polymers-14-04670-f005:**
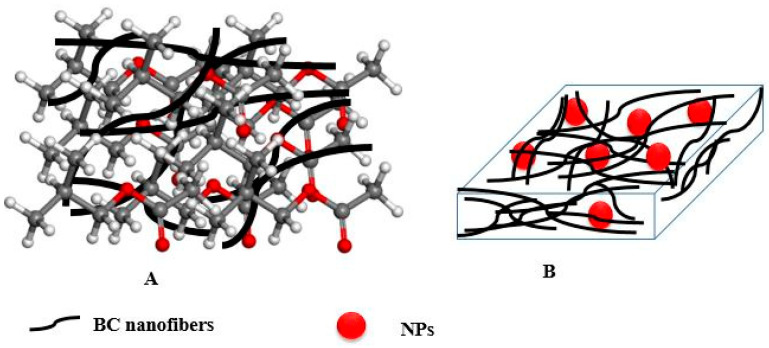
The use of BC as a reinforcement (**A**) or matrix (**B**) to prepare nanocomposites.

**Figure 6 polymers-14-04670-f006:**
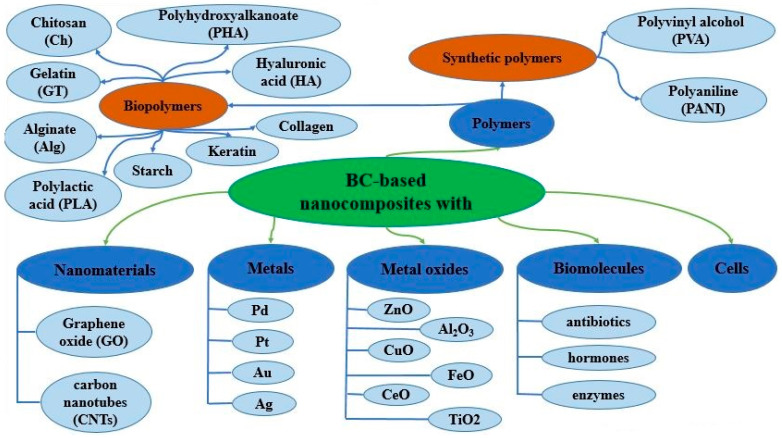
A generalized classification of BC composites prepared with various materials of organic and inorganic nature.

**Figure 7 polymers-14-04670-f007:**
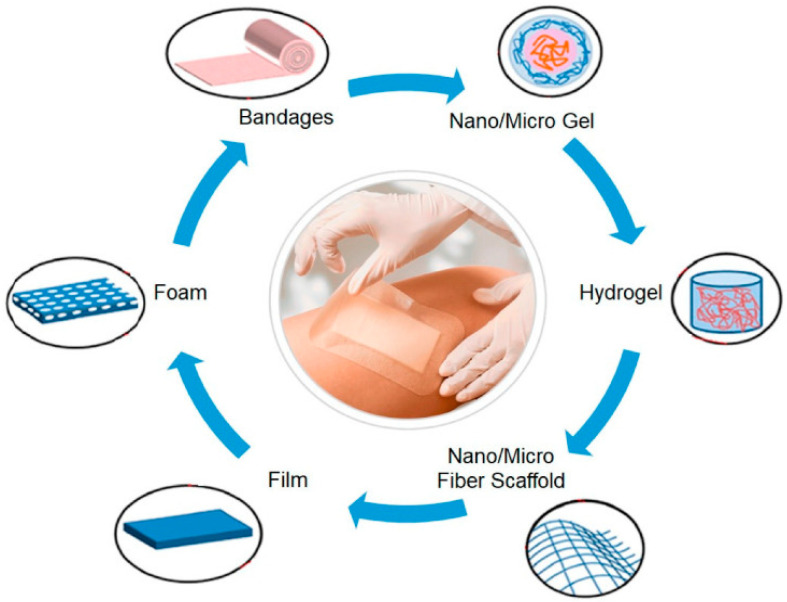
Different forms of wound dressings. Reprinted with permission from Ref. [185].

**Figure 8 polymers-14-04670-f008:**
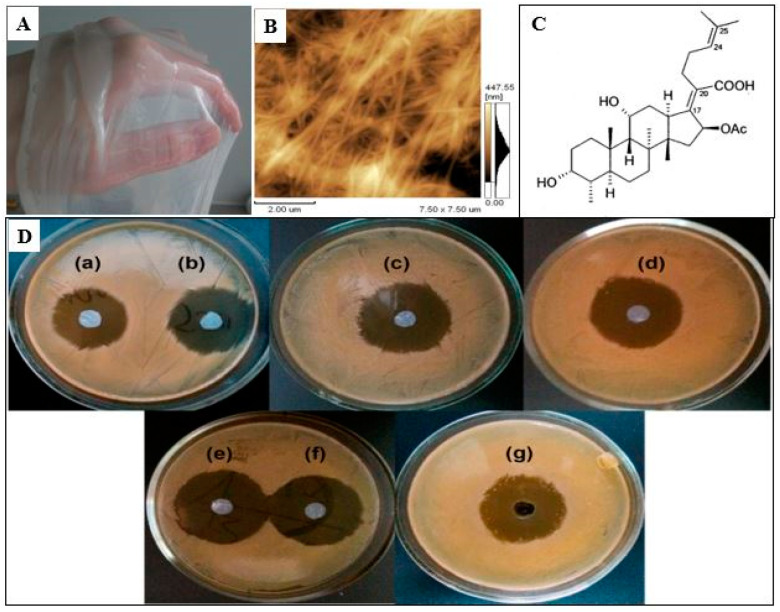
BC gel film (**A**); AFM image of the BNC (**B**); chemical structure of fusidic acid (**C**); antimicrobial activity of BNC/ FA composites (**D**) against *S. aureus* after 24 h of exposure: BNC_0.1_ (a); BNC_0.2_ (b); BNC_0.3_ (c); BNC_0.4_ (d); after 1 h of exposure BNC_0.4_ (e); after 2 h of exposure BNC_0.4_ (f); control FA _0.4_ (g). Adapted with permission from Ref. [213].

**Figure 9 polymers-14-04670-f009:**
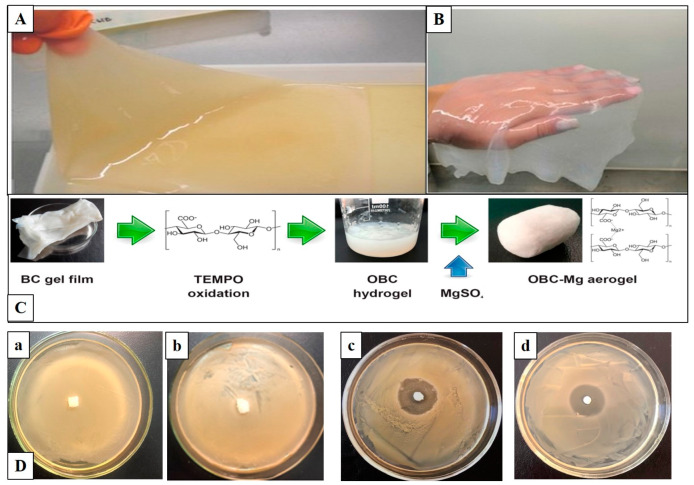
A BC gel film on the surface of the HS medium (**A**), after its purification (**B**), scheme for producing NBC or OBC aerogels (**C**), and antimicrobial activities of aerogels against *S. aureus* (**D**): NBC (a), OBC (b), biocomposite NBC/SF_100_ (c), biocomposite OBC/SF_100_ (d). Adapted with permission from Ref. [214].

**Figure 10 polymers-14-04670-f010:**
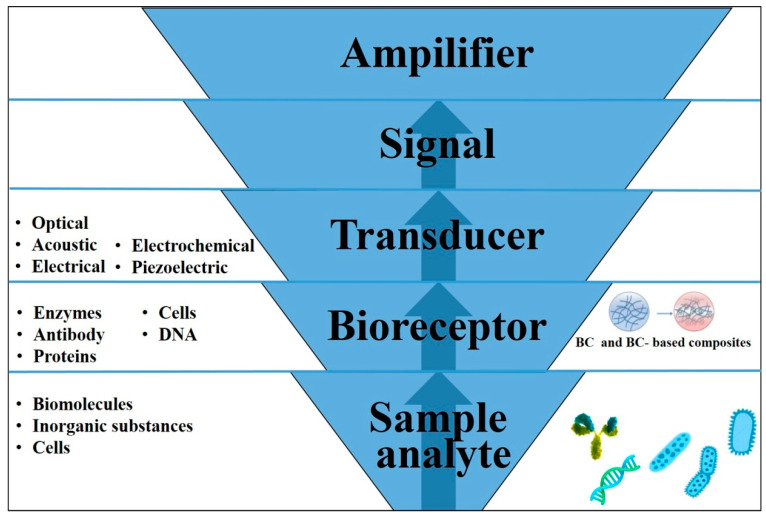
Schematic diagram representing the different components of a biosensor. The interaction between the sample analyte (biomolecules, inorganic substances, cells) and bioreceptor (the recognition molecule) is captured as a signal by the transducer which is then used for detection through signal transduction (optical, acoustic, electrochemical, piezoelectric).

**Figure 11 polymers-14-04670-f011:**
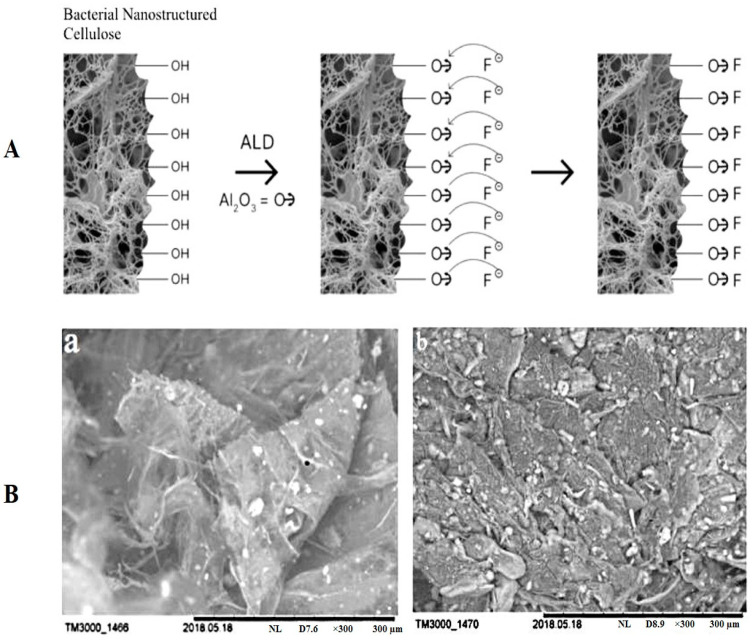
The scheme of the preparation process of biocomposites based on BC for the adsorption of fluoride (**A**) and (a) SEM image of BC at 300 magnification; (b) SEM image of BC modified with aluminum oxide at 300 magnification (**B**). Adapted with permission from Ref. [130].

**Table 1 polymers-14-04670-t001:** Sources of isolation BC-producing strains.

Source of Isolation	Strain	Reference
Kombucha	*K. xylinus* B-12068	[44]
	*G. hansenii* GH-1/2008 (B-10547)	[45]
	*K. hansenii* JR-02	[53]
	*K. hansenii* SI1	[54]
	*K. hansenii* LMG 23726	[55]
	*K. rhaeticus* P 1463	[45]
	*K. intermedius* AF2	[46]
	*K. rhaeticus* K3	[56]
	*K. sucrofermentans* B-11267	[39]
Vinegar	*K. hansenii* DSM 5602^T^	[55]
	*K. medellinensis* LMG 1693^T^	[47]
Apple cider vinegar	*K. europaeus* LMG 20956 *K. melaceti* AV382^T^ *K. melomenusus* AV436^T^ *K. melomenusus* SI3083 *K. nataicola* LMG 1536^T^ *K. oboediens* AV371 *K. oboediens* BJK_8C *K. oboediens* SI3053 *K. pomaceti* T5K1^T^ *K. pomaceti* AV445 *K. pomaceti* AV446 *K. pomaceti* SI3133 *K. saccharivorans* AV378 *K. saccharivorans* JK_3A *K. swingsii* LMG 22125^T^ *G. entanii* SI2035 *G. entanii* AV429	[55]
Red wine vinegar	*K. europaeus* LMG 18494	[55]
Fruit	*K. maltaceti* SKU 1109	[55]
	*Gluconacetobacter* sp. F6	[52]
Rotten fruits	*G. xylinus* BCZM sp.	[57]
Rotten banana Kombucha	*Komagataeibacter* sp. CCUG73629 *Komagataeibacter* sp. CCUG73630	[58]
Fruit juice	*K. intermedius*	[49]
	*G. swingsii, G. rhaeticus*	[50]
	*Gluconacetobacter* sp. gel_SEA623-2	[51]
Organic apple juice	*K. rhaeticus* DSM 16663^T^ *K. swingsii* LMG 22125^T^	[55]
Beet juice	*K. saccharivorans* LMG 1582^T^	[55]
Nata de coco	*K. nataicola* LMG 1536^T^	[55]
Honey wine	*K. maltaceti* P285	[59]
Coconut milk	*K. cocois* sp. nov.	[60]
Tibicos symbiotic community	*K. hansenii* B-12950	[39]

**Table 2 polymers-14-04670-t002:** A summary of developing bacterial cellulose-based composites by different strategies for diverse biomedical applications.

Applications	Reinforcement Material	Synthesis Approach	Enhanced/Imparted Features	Refs.
Tissue engineering	Poly(pyrrole) and carbon nanotubes	Solvent dissolution and regeneration	Thermal and mechanical stability, recoverability, swelling behavior, electrical conductivity, biocompatibility	[148]
	κ-carrageenan	In situ impregnation	Mechanical strength, water holding and controlled release, swelling behavior, biocompatibility, gene expression	[149]
	Sodium chloride crystals	Solvent dissolution and regeneration	Porosity, 3D morphology, biocompatibility, 3D cell growth	[150]
	Graphene oxide/ reduced graphene oxide	Ex situ addition	Mechanical strength, hydrophilicity, biocompatibility, electrical conductivity	[151,152]
	Quaternized chitosan	Ex situ addition	Porosity, water holding and control release, thermal stability, cytocompatibility, antibacterial activity	[153]
	ZnO nanoparticles	Solvent dissolution and regeneration/Ex situ addition	Thermal and mechanical strength, antibacterial activity, biocompatibility	[154,155]
	Plant extract	Ex situ addition	Mechanical strength, water uptake and controlled release, antibacterial activity, biocompatibility	[141]
	Titanium dioxide nanoparticles	In situ impregnation and regeneration, cell-free synthesis	Thermal and mechanical strength, uniform distribution of nanoparticles, antibacterial activity, biocompatibility	[156,157]
Wound dressing, healing, and hemostasis	Collagen	In situ impregnation	Thermal and mechanical stability, cytocompatibility, collagen synthesis	[158]
	Gelatin	Ex situ addition and physical stretching	Electric field stimulation, aligned fibers, in vitro and in vivo biocompatibility, wound closure, formation of granulation tissues, collagen synthesis, angiogenesis, gene expression	[159]
	Poly(vinyl alcohol), silk sericin, azithromycin	Ex situ addition	Mechanical strength, porosity, anti-inflammation, antibacterial activity, in vitro and in vivo biocompatibility, successful treatment of chronic wound biofilms	[160]
	Chitosan and diamond nanoparticles	Ex situ addition	Thermal and mechanical stability, electrical modulus, antibacterial activity, biocompatibility	[161]
	Gelatin and selenium nanoparticles	In situ impregnation	Mechanical and tensile strength, antioxidant and anti-inflammatory properties, antibacterial activity, angiogenesis, collagen synthesis, gene expression, granulation tissue formation	[162]
	Chitosan and collagen	Ex situ addition	Mechanical stability, biocompatibility, antimicrobial activity, in vitro and in vivo biodegradation, hemostasis	[163]
	Poly (2-hydroxyethyl methacrylate) and silver nanoparticle	Ex situ addition	Thermal and mechanical stability, optical transparency, antibacterial activity, biocompatibility	[164]
	Ag nanoparticles	In situ impregnation	Mechanical strength, antibacterial activity, biocompatibility, collagen synthesis, in vivo burn wound healing, re-epithelization, expression of inflammatory, angiogenesis, and growth factor genes, successful third-degree burn wound healing	[165]
	Montmorillonite	Ex situ addition	Mechanical strength, antibacterial activity, water holding and controlled release rate, biocompatibility	[166,167]
	Curcumin	Ex situ addition	Mechanical strength, antibacterial activity, reepithelization, vascularization, wound closure, successful partial-thickness skin burns in animal model	[168]
Bone tissue engineering	Cellulose nanocrystals and protein	Chemical modification	Mechanical strength, thermal stability, morphology, biocompatibility	[169]
	Otoliths and collagen	Post-synthesis loading	Osteoblast activity, regularity, osteo-reabsorption activities	[170]
	Col_1_	Post-synthesis cross-linking	Low tensile strength and elastic modulus, high strain, regular cell growth, biocompatibility, non-toxicity	[171]
	Hydroxyapatite and carboxymethyl cellulose	Ex situ addition	Mechanical strength, thermal stability, biocompatibility	[172]
Drug delivery	Graphene oxide and ibuprofen	Ex situ addition	Controlled in vitro drug release, biocompatibility, electrical conductivity, tensile strength	[173]
	Poly(ethylene imine)	Ex situ addition	Improved morphology, adsorption, controlled in vitro drug release, biocompatibility	[174]
	Polyaniline	Ex situ addition	Electrical conductivity, pH-responsiveness, sustained in vitro drug release	[175]
	---	Freeze-drying	pH-dependent drug release, 3D morphology, porous structure	[176]
Biosensors, bioelectronics, and diagnosis	Polyaniline and carbon nanotubes	Ex situ addition	Porous morphology, thermal stability, electrical conductivity	[177]
	Carbon nanotubes and poly(ethylene imine)	Ex situ addition	High density phage immobilization, mechanical stability, surface charge, electrical conductivity, antibacterial activity, stability and reuse of sensing interface, bacterial detection with high specificity	[178]
	Chitosan	Ex situ addition	Mechanical stability, water uptake and controlled release, biocompatibility, 3D cell growth, use for diagnosis of ovarian cancer	[179]
Artificial blood vessels	Poly(dimethyl siloxane)	Molding	Patterned morphology, mechanical strength, tubular shape, biocompatibility, non-toxicity	[180]
Heart valve	Poly(vinyl alcohol)	Ex situ addition	Tensile strength and elastic modulus, anisotropy, optical transparency, biocompatibility	[181]
Artificial cornea	Poly(vinyl alcohol)	Ex situ addition	Optical transparency, mechanical strength, thermal stability, biocompatibility	[182]
Artificial kidney and liver	---	3D printing	Biocompatibility, mechanical strength, porous morphology	[183]
Neural tissue regeneration	Agarose	Molding	Aligned fibers, mechanical strength, biocompatibility	[184]

## Data Availability

Not applicable.

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
