# Peer review of "Bacterial Cellulose-Based Polymer Nanocomposites: A Review"

_polymers, 2022, doi:10.3390/polym14214670_

Round 1

Reviewer 1 Report

Present manuscript deal with Bacterial cellulose and derived nanocomposites for biomedical and biotechnology applications. It is an interesting contribution, but need major changes before publication

1-Reference checking: authors mix numbered and name citations. This should be revised in the text

2-English structure should be improved. For instance, several parts include too short paragraphs not adequately connected to the text

3-Authors should use figures with higher quality and definition. For instance Figures  Figure 4 C (XRD pattern) and Figure 12, should be replaced for better ones

4-Biosensor and adsorbents sections in particularly sections are poorly witten and should be carefully revised. A more complete discussion on the advantages of BC for such applications should be presented. Particularly for adsorbet section I reccomend inclusion of (photo)catalytic BC based materials for water contaminants removal

Author Response

Response to Reviewer 1 Comments

Thank you very much for all of your detailed comments and suggestions. We found them quite useful as we approached our revision. The authors completely agree with reviewers comment.

Point 1: Reference checking: authors mix numbered and name citations. This should be revised in the text.

Response 1: Thank you very much for your correction.The text was revised.

Point 2. English structure should be improved. For instance, several parts include too short paragraphs not adequately connected to the text.

Response 2: The whole manuscript was revised.

Point 3. Authors should use figures with higher quality and definition. For instance Figures  Figure 4 C (XRD pattern) and Figure 12, should be replaced for better ones

Response 3: Thank you very much for your recommendation.The quality of Figure 4 and Figure 11 (12) was improved.

Point 4. Biosensor and adsorbents sections in particularly sections are poorly witten and should be carefully revised. A more complete discussion on the advantages of BC for such applications should be presented. Particularly for adsorbet section I reccomend inclusion of (photo)catalytic BC based materials for water contaminants removal.

Response 4: Thank you very much for your recommendation. Biosensor and adsorbents sections were revised. We have added information on photocatalysis as another promising method for the degradation of organic pollutants and dyes in wastewater.

Reviewer 2 Report

In the introduction, the first sentence is too weak to be a stand-alone paragraph. Please check.

Please expand the conclusion by emphasizing the interesting achievements of this study.

Author Response

Response to Reviewer 2 Comments

Thank you very much for all of your detailed comments and suggestions. We found them quite useful as we approached our revision. The authors completely agree with reviewers comment.

Point 1: In the introduction, the first sentence is too weak to be a stand-alone paragraph. Please check.

Response 1: Thank you very much for your correction.The first sentence was merged with the following.

Point 1: Please expand the conclusion by emphasizing the interesting achievements of this study.

Response 2: RESPONSE: Thank you very much for your recommendation. We rewrote the conclusion by emphasizing the interesting achievements of this study.

Reviewer 3 Report

Dear Author,

You took nice efforts but I do not see any novelty in this review so suggested a major revision. Patents and marketed technology are missing. Need to find novelty aspects to stand your article different as compared to other similar kinds available in the scientific community.I suggest major revision and rewritting.

1. Figure 4. and Figure 12. Does image quality need to improve?

2. Abstract need to rewrite?

3. Conclusion need to revise carefully?

4. Table for Biomedical application literature missing and needs to rewrite and revise?

5.Future perspective need to add in mnuscript ?

Author Response

Response to Reviewer 3 Comments

Thank you very much for all of your detailed comments and suggestions. We found them quite useful as we approached our revision. The authors completely agree with reviewers comment.

Point 1: Figure 4. and Figure 12. Does image quality need to improve?

Response 1: Thank you very much for your correction.The quality of Figure 4 and Figure 11 (12) was improved.

Point 2: Abstract need to rewrite

Response 2: Thank you very much for your recommendation. Abstract was rewrote.

Point 3: Conclusion need to revise carefully.

Response 3: Thank you very much for your recommendation.We rewrote the conclusion by emphasizing the interesting achievements of this study.

Point 4: Table for Biomedical application literature missing and needs to rewrite and revise?

Response 4: Thank you very much for your recommendation. The Table «A summary of developing bacterial cellulose-based composites by different strategies for diverse biomedical applications» is presented in improved manuscript (Table 2).

Response 5: Future perspective need to add in manuscript.

RESPONSE: Thank you very much for your recommendation. We added future perspectives in the conclusion.

Reviewer 4 Report

Abstract -> add scope of the review. The reader should get whole picture of the manuscript.

Introduction -> First paragraph contains only 1 sentence. This should be revised.

The length of the paragraph is too short. Revise for the whole manuscript e.g. P 1, P2, …

There is mixture of citation style. Please recheck e.g. P2

Recheck the use of bacterial cellulose VS BC. Full and abbreviation are mixed. There is no need to mention full name again, once you write the BC.

Table 1 -> Recheck the writing style for scientific name. Should they be italics?

Fig. 10 is difficult to understand. Please consider revising the figure or add more explanation in the figure caption.

P23 Add more discussion e.g., These biopolymers also had high potential to produce sustainable materials for packaging applications. (Polymers, 14(18), 3706;  Polymers, 14(18), 3793). Development of active nanocomposite polymers e.g., incorporating ZnO nanoparticles has been proved to provide extra antimicrobial efficiency which preserved quality of packaged products (Colloids and Surfaces B: Biointerfaces, 214, 112472; Polymers, 14(19), 4042).

Fig. 12 -> Improve resolution

Conclusions -> Please add challenges or future perspective.

Author Response

Response to Reviewer 4 Comments

Thank you very much for all of your detailed comments and suggestions. We found them quite useful as we approached our revision.

Point 1: Introduction -> First paragraph contains only 1 sentence. This should be revised.

Response 1: Thank you very much for your correction.The first sentence was merged with the following.

Point 2: The length of the paragraph is too short. Revise for the whole manuscript e.g. P 1, P2,

Response 2: The whole manuscript was revised.

Point 3: There is mixture of citation style. Please recheck e.g. P2

Response 3: Thank you very much for your correction.The manuscript was revised.

Point 4: Recheck the use of bacterial cellulose VS BC. Full and abbreviation are mixed. There is no need to mention full name again, once you write the BC.

Response 4: Thank you very much for your correction.The manuscript was revised.

Point 5: Table 1 -> Recheck the writing style for scientific name. Should they be italics?

Response 5: Microorganisms name should be in italics forma.

Point 6: Fig. 10 is difficult to understand. Please consider revising the figure or add more explanation in the figure caption.

Response 6: Thank you very much for your recommendation. We have added more explanation in the figure caption.

Point 7: P23 Add more discussion e.g., These biopolymers also had high potential to produce sustainable materials for packaging applications. (Polymers, 14(18), 3706;  Polymers, 14(18), 3793). Development of active nanocomposite polymers e.g., incorporating ZnO nanoparticles has been proved to provide extra antimicrobial efficiency which preserved quality of packaged products (Colloids and Surfaces B: Biointerfaces, 214, 112472; Polymers, 14(19), 4042).

Response 7: Sorry, but we didn't understand the question. The review discusses BC-based nanocomposites for biomedical applications including wound healing, drug delivery, and tissue engineering.

Point 8: Fig. 12 -> Improve resolution.

Response 8: Thank you very much for your recommendation. Figure 11 (12) resolution was improved.

Point 9: Conclusions -> Please add challenges or future perspective.

Response 9: Thank you very much for your recommendation. We added future perspectives in the conclusion

Round 2

Reviewer 1 Report

Revised manuscript may be accept after re-checking english again. For instance change "airgel" to "aerogel". Also several paragrahs are still too short. Check please

Author Response

Response to Reviewer 1 Comments

 Thank you very much for all of your comments and suggestions. We found them quite useful as we approached our revision. The authors completely agree with reviewers comment.

Point 1: Revised manuscript may be accept after re-checking english again. For instance change "airgel" to "aerogel".

Response 1: Thank you very much for your recommendation and correction. We re-checked the English and made changes to the text.

Point 2:  Also several paragrahs are still too short. Check please.

Response 2: Thank you very much for your recommendation. We increased the length of the paragraphs.

Reviewer 3 Report

Dear Author Team,

Now your article looks promising, can you please check sentence alignment and resubmit it.

Author Response

Response to Reviewer 3 Comments

Thank you very much for all of your comments and suggestions. We found them quite useful as we approached our revision. The authors completely agree with reviewers comment.

Point 1: Now your article looks promising, can you please check sentence alignment and resubmit it.

Response 1: Thank you very much for your recommendation. We checked the alignment of sentences and made changes to the text.